# A New Approach for Phage Cocktail Design in the Example of Anti-Mastitis Solution

**DOI:** 10.3390/pathogens13100839

**Published:** 2024-09-27

**Authors:** Daria Królikowska, Marta Szymańska, Marta Krzyżaniak, Arkadiusz Guziński, Rafał Matusiak, Agnieszka Kajdanek, Edyta Kaczorek-Łukowska, Agnieszka Maszewska, Ewelina A. Wójcik, Jarosław Dastych

**Affiliations:** 1Proteon Pharmaceuticals, Tylna 3a, 90-364 Łódź, Poland; mszymanska1@proteonpharma.com (M.S.); mkrzyzaniak@proteonpharma.com (M.K.); aguzinski@proteonpharma.com (A.G.); rmatusiak@proteonpharma.com (R.M.); akajdanek1@proteonpharma.com (A.K.); amaszewska@proteonpharma.com (A.M.); ewojcik@proteonpharma.com (E.A.W.); jdastych@proteonpharma.com (J.D.); 2Department of Microbiology and Clinical Immunology, Faculty of Veterinary Medicine, University of Warmia and Mazury in Olsztyn, Oczapowskiego 13, 10-719 Olsztyn, Poland; edyta.kaczorek@uwm.edu.pl

**Keywords:** bacteriophage therapy, bioinformatics-based design, structural analyses of phages, antibiotic resistance, mastitis, biofilm, milk model

## Abstract

The studies on phage therapy have shown an overall protective effect of phages in bacterial infections, thus providing an optimistic outlook on the future benefits of phage-based technologies for treating bacterial diseases. However, the therapeutic effect is highly affected by the proper composition of phage cocktails. The rational approach to the design of bacteriophage cocktails, which is the subject of this study, allowed for development of an effective anti-mastitis solution, composed of virulent bacteriophages acting on *Escherichia coli* and *Staphylococcus aureus*. Based on the in-depth bioinformatic characterization of bacteriophages and their in vitro evaluation, the cocktail of five phages against *E. coli* and three against *S. aureus* strains was composed. Its testing in the milk model experiment revealed a reduction in the number of *S. aureus* of 45% and 30% for *E. coli* strains, and in the study of biofilm prevention, it demonstrated 99% inhibition of biofilm formation for all tested *S. aureus* strains and a minimum of 50% for 50% of *E. coli* strains. Such insights justify the need for rational design of cocktails for phage therapy and indicate the potential of the developed cocktail in the treatment of diseased animals, but this requires further investigations to evaluate its in vivo efficacy.

## 1. Introduction

Mastitis is an intramammary inflammation, and is the most common and the most expensive disease concerning dairy animals worldwide [1]. The disease negatively affects udders, the quality and quantity of milk, and the general welfare of animals, subsequently increasing rearing and prevention costs [2]. Mastitis-affected milk was shown to contain a variety of pathogenic bacteria, toxins, increased somatic cell counts (SCCs) correlated with depleted fat, lactose, solid not fat (SNF), and ash content, but was also shown to have lumps and undesirable color change [1,2,3]. Milk with altered chemical and physical properties that make it improper for human consumption, the costs of drugs and the management of the disease, mortality, and poor animal welfare ultimately result in enormous financial losses [1].

Depending on the origin of the pathogen, mastitis can be classified into two types: environmental or contagious. The former is caused by bacteria spreading in the bovine-related environment, particularly the milking hall, e.g., soil, feces, stagnant water, and bedding material. On the contrary, contagious mastitis is caused by bacteria that are transferred from infected to healthy animals usually at the time of milking through the milker’s hands, milking equipment, or hygiene products such as towels. The most frequently mentioned environmental pathogens are *Escherichia coli* and *Streptococcus uberis*, whereas the most common causatives of contagious mastitis include *Staphylococcus aureus* and *Streptococcus agalactiae* [4,5]. Moreover, mastitis can be distinguished either as clinical or subclinical, with the latter mainly associated with the contagious type of mastitis [1,6]. Subclinical mastitis is correlated with a significant increase in IgG antibodies, Cl, Na, and free fatty acids (FFA), reduced lactose and total protein count, but also increased pH value of mastitis milk that is associated with the severity of inflammatory process. The peculiarity of subclinical mastitis is greatly reduced milk production with a lack of noticeable milk or udder abnormalities that make the bovine infection not detectable, which eventually affects the consumers and causes more than three to four times more losses of milk production than clinical mastitis [6,7].

Mastitis infections are the main reason for the usage of antibiotics in the dairy industry. Unfortunately, lack of targeted treatment and the overuse and misuse of antibiotics lead to the emergence of bacterial antibiotic resistance, the non-responsiveness of cattle, and increased risk of entrance of resistant bacteria into the food chain [1,8]. Kovačević Z. et al. [9] indicated a direct association between the use of common antibiotics in mastitis treatment and antimicrobial resistance of mastitis-associated pathogens isolated from milk samples. The World Health Organization (WHO) declared antibiotic resistance one of the biggest threats to global health and food security and pointed to the urgent need for development of alternative therapies against bacteria [10,11]. Hence, nowadays, alternatives to the existing therapies with antibiotics and preventive approaches are of special interest. Among antimicrobial treatments with promising therapeutic efficacy, there are herbal medicines based on essential oils (EOs) [12], recombinant peptides [13], bacteriocins [14], and others. This study focused on one feasible perspective: the employment of bacteriophages.

Bacteriophages, or phages, are viruses that specifically infect bacteria. Their number is estimated at approximately 10^31^ units, which is ten times greater than the number of bacteria and makes them the most numerous beings in our biosphere [15]. Compared to antibiotics, their greatest advantage is the high specificity of action; phages infect selected species of bacteria or, even more precisely, only some strains within a species. Other advantageous qualities of phage therapy include a ubiquitous nature and low natural phage toxicity [16]. Phage therapy is typically implemented as cocktails composed of a variety of phages that should cover a wide range of pathogenic strains [17]. However, when composing effective phage cocktails, the greatest challenge is the appropriate composition of the phage cocktail. The current work aimed to develop a rational approach to the composition of effective phage cocktails, in which in silico analysis supplemented with laboratory results plays a critical role. In silico analysis focused on the assessment of phage-host interactions by precisely identifying receptors allowing for the selection of phages recognizing different structures on the bacteria’s cell surface. Additionally, this approach enables the avoidance or minimization of phage resistance in bacteria, finally giving the bacteriophage cocktail usefulness in the prevention or treatment of mastitis infections in dairy cattle caused by *E. coli* and *S. aureus*. The approach proposed in the study constitutes a new direction in designing phage cocktails.

## 2. Materials and Methods

### 2.1. Bacterial Collection and Growth Conditions

A collection of bacterial pathogenic strains (also referred to as basic collection) isolated from cows with mastitis symptoms, obtained from the northeast region of Poland at the University of Warmia and Mazury (UWM, Olsztyn, Poland), including 18 *E. coli* and 15 *S. aureus* unique strains was used in this study (the property of Proteon Pharmaceuticals S. A.). The strains are listed in Table 1. For short, only the initial number of the strain can be used, e.g., *E. coli* 090 instead of *E. coli* 090PP2016. *S. aureus* strains have been previously well-characterized [18]. Bacterial aliquots were stored as glycerol stocks at −80 °C, and then *E. coli* strains were inoculated on Luria-Bertani agar plates (LB; BIOMAXIMA, Lublin, Poland) and *S. aureus* strains were inoculated on Tryptic Soy Broth agar plates (TSB; BIOMAXIMA, Lublin, Poland). After overnight incubation (18–20 h) at 37 °C, three colonies were suspended in a respective liquid medium and incubated with shaking (140 rpm) at 37 °C for 2–3 h unless otherwise stated in the experiment description.

### 2.2. Bacteriophages

This study involved the examination of 5 *E. coli*-specific and 3 *S. aureus*-specific bacteriophages (Table 2), isolated from environmental samples using the double-layer agar method [19]. In brief, 100 µL of an environmental sample in LB/TSB medium (depending on strain, LB for *E. coli*, TSB for *S. aureus*) was added to the test tube with 4 mL of molten agar (0.7%) and 100 uL of overnight bacterial culture. The tube was mixed at the vortex, and the content was poured over the surface of the plate with solid agar medium (2% agar LB/TSB medium). The plate was left to dry and then incubated at 37 °C for 24 h. After incubation, the plaques were visible if phages were present in the environmental sample.

A bacteriophage cocktail at the titer of 2 × 10^8^ PFU/mL (plaque forming unit/milliliter) was used in the studies. Each bacteriophage included in the cocktail was amplified in a separate culture. In this procedure, the culture medium was inoculated with the host production strain and incubated at 37 °C for approximately 2 h. Next, the selected phage was added to the bacterial culture, and incubation proceeded for an additional 3 h at 37 °C, leading to cell lysis. This process enables obtaining a high titer of amplified bacteriophage at a level of about 1 × 10^9^ PFU/mL. After amplification, the biomass was separated from the phage-containing culture fluid by membrane filtration with a 0.2 µm pore size. Once the amplification procedure was completed, the titer of the bacteriophage was assessed by phage enumeration using a double agar overlay plaque assay [19]. In brief, 100 µL of 10-fold dilutions of phages in LB/TSB medium were added to the test tube with 4 mL of molten agar (0.7%) and 100 uL of overnight bacterial culture. The tube was mixed at the vortex, and the content was poured over the surface of the plate with solid agar medium (2% agar LB/TSB medium). The plate was left to dry and then incubated at 37 °C for 24 h. The phage titer was determined by the following calculation:$$M_{f}=\frac{\sum c}{V\cdot d\cdot (N_{1}+0.5N_{2}+0.25N_{3}+\cdots )}$$
where:
$M_{f}$—bacteriophage titer (PFU/mL),c—the sum of plaques on all plates for a given repetition,V—volume of inoculated phage lysate (0.1 mL),$N_{1}$—number of plates from 1st dilution,$N_{2}$—number of plates from 2nd dilution,$N_{3}$—number of plates from 3rd dilution,d—the lowest dilution used.

Each bacteriophage titer was calculated for a mixture in suspensions at equal titer value. Finally, the preparation was completed with SM buffer (50 mM Tris-Cl, pH 7.5, 99 mM NaCl, 8 mM MgSO_4_, 0.01% gelatin) to obtain an anti-*E. coli* component at a titer of 1 × 10^8^ PFU/mL (consisting of 5 bacteriophages, each at a titer of 2 × 10^7^ PFU/mL) and an anti-*S. aureus* component at a titer of 1 × 10^8^ PFU/mL (including 3 bacteriophages, each at a titer of 3 × 10^7^ PFU/mL), and the microbiological sterility of the product was analyzed with the usage of AlamarBlue^®^ (BioRad, Hercules, CA, USA). For this purpose, 50 µL of AlamarBlue^®^ reagent were added to two 1.5 mL tubes. An amount of 100 µL of the sample tested was added to one tube (the second tube is a control sample). Both tubes were incubated for 24 ± 3 h at 37 °C in the dark. After incubation, the test result was correct when no color change was observed (navy blue), which indicates the lack of live bacterial cells in the product.

### 2.3. Characterization of Bacteriophages

For the characterization of isolated bacteriophages, they were subjected to a series of analyses, starting with genetic material isolation according to the modified method of Su et al. [20]. Next, the genomes of bacteriophages were sequenced with the Next Generation Sequencing (NGS) method on the Illumina platform by an external provider that used the Nextera method for library construction. The draft genomes were assembled de novo with SPAdes v3.9.1 [21], and the obtained sequences were structurally annotated, applying PROKKA v1.3.1 [22]. Functional annotations were made using the programs InterProScan v 5.62-94.0 [23], HmmScan v3.3.2 [24], HHpred (Online access 11 April 2023: https://toolkit.tuebingen.mpg.de/tools/hhpred) [25], PhaNNs v0.3 [26], and BLASTp v2.9.0 [27]. The annotated phage’s structural proteins that build the bacteriophage receptor recognition systems were deeply analyzed by BLASTp and HHblits v3.3.0 [28] for detection of native oligomeric state by homology analysis of proteins with a known native oligomeric state deposited in the Protein Data Bank (online access: 14 January–11 November 2022). The detected structural proteins were analyzed using an internal pipeline (unpublished data). The first step of this analysis was the estimation of the sequence similarity and protein models’ similarity to known bacteriophage structural proteins deposited in the Protein Data Bank. The second step was the digital reconstruction of functional elements of the virion, like the head fibers, tail oligomers, tail-fibers elements, and puncturing devices, using AlphaFold-Multimer [29,30,31] and Swiss-Model [32]. These analyses allowed for the detection of receptor binding proteins (RBPs), which were later bioinformatically analyzed in depth for the detection of crucial domains and amino acids motifs involved in recognition of the bacteriophage receptors. The performed bioinformatic analysis allowed for the prediction of the phage receptors. The taxonomy classification was made by VIRIDIC v1.0 [33]. Phage’s genome maps were generated by SnapGene Viewer v7.2.1 (SnapGene software, www.snapgene.com). Phage’s genome comparisons were prepared in Easyfig 2.2.5 [34].

Finally, the morphology of each bacteriophage was determined by using a JEOL 1010 transmission electron microscopy TEM (JOEL Ltd., Tokyo, Japan) in the Laboratory of Microscopic Imaging and Specialized Biological Techniques of the Faculty of Biology and Environmental Protection, University of Łódź, Poland. Bacteriophage lysates were centrifuged at 24,500× *g* for 3.5 h at 4 °C. The precipitates were suspended twice in a 5% ammonium molybdate solution (Signa-Aldrich, St. Louis, MI, USA) using the same spin parameters. Thus, obtained sediments were suspended in 5% ammonium molybdate to reach the final titer of 1 × 10^11^ PFU/mL. A drop of such phage suspension was deposited on a formvar-coated and carbon-sprayed 200 mesh copper grid (Polysciences Inc., Warrington, PA, USA) and stained for 45 s with 2% phosphotungstic acid in darkness.

### 2.4. Lytic Activity

For the assessment of the lytic activity of selected bacteriophages, 18–20 h bacterial cultures (2 × 10^8^ CFU/mL, colony forming unit/milliliter) were 100-fold diluted with LB or TSB medium depending on strain and applied to four wells in a 96-well plate in a volume of 100 µL per well. Two wells were supplemented with 20 µL of tested bacteriophage lysates (titer 2 × 10^8^ PFU/mL, MOI 20), while the remaining two wells were positive controls (with 20 µL of SM buffer, pH = 7.5). The next four wells were filled with 100 µL of medium per well (LB medium for *E. coli*, TSB for *S. aureus)*, of which 20 µL of phage lysate was added to two of them as negative controls, and the remaining two wells were added with 20 µL of buffer. Plates prepared in this manner were placed in a Tecan Sunrise Basic Microplate Reader, and the absorbance of samples (optical density at wavelength 620 nm, OD_620_) was measured every 20 min, at 37 °C, for 280 min. Based on analogous absorbance measurement results for avian pathogenic *E. coli* (APEC), the following assumption on the degree of the bacterial growth inhibition was made: the difference between the positive control OD and the test sample OD above 0.1 indicates strong inhibition of growth by the tested phage; the difference between 0.1 and 0.05 indicates weak inhibition; and the difference below 0.05 is considered as no influence of bacteriophage on the bacteria growth. The evaluation of the degree of the bacterial growth inhibition was analyzed previously in Proteon on taxonomically different bacteriophages and a large number of bacterial strains. 

### 2.5. Induction of Phage Resistance

Bacterial host strains were examined for potency to gain resistance to chosen phages. An amount of 100 µL of phage lysates, each in a titer of 1 × 10^9^ PFU/mL, were added to separate Eppendorf tubes. Next, 100 µL of 100-fold diluted 18–20 h bacterial cultures with a density around 1 × 10^6^ CFU/mL (OD_600_ = 0.5) were then added to each Eppendorf tube. The control sample contained 100 µL of growth medium (LB for *E. coli*, TSB for *S. aureus*) and 100 µL of diluted bacterial culture. After 10 min of incubation at 37 °C, 100 µL of each sample was collected, spread onto the double-layer agar plates, and incubated for 24 h at 37 °C. The double agar layer plates intended for the isolation of resistant bacteria were prepared by adding 100 µL of phage lysates to the top agar, while the top agar of plates for control samples contained 100 µL of medium in which phage lysates were suspended. Resistant colonies observed on plates with phages in top agar were re-cultured on a liquid and solid medium. Subsequently, their resistance to selected bacteriophages was evaluated with a spot test, together with the assessment of their sensitivity to the remaining phages chosen for the cocktail composition. Obtained resistant strains were preserved as glycerol stocks and also sequenced by the illumina NGS platform and deeply analyzed to determine the mutated genes that induce the resistance phenotype. The mutations were detected by Snippy v4.6.0 software [35]. The detected mutations were analyzed for assessment of their impact on the protein function, which allows for the description of the resistance-gaining mechanism by bacteria exposed to analyzed phages.

### 2.6. Bacteriophage Cocktail Host Range (Specificity)

To investigate the host range of bacteriophage cocktail, a serial dilutions spot test method [36] on 17 *E. coli* and 15 *S. aureus* strains isolated from cows with mastitis was performed. The bacteriophage cocktail (2 × 10^8^ PFU/mL) was ten-fold serial diluted in SM buffer (2 × 10^8^ PFU/mL, 2 × 10^7^ PFU/mL, 2 × 10^6^ PFU/mL, 2 × 10^5^ PFU/mL, 2 × 10^4^ PFU/mL, 2 × 10^3^ PFU/mL). Next, 100 µL of overnight bacterial suspension (2 × 10^8^ CFU/mL) were added to test tubes containing molten soft agar (agar 0.7%; LB + 5% glycerol for *E. coli* and TSB for *S. aureus*), mixed and plated on Petri dishes with solid agar medium (agar 2%; LB + 5% glycerol for *E. coli* and TSB for *S. aureus*) and left to dry. Then, 10 µL of each dilution were spotted onto the plate and left to dry. SM buffer was used as a negative control. Plates were incubated at 37 °C for 18–24 h. Determination of bacteria sensitivity to bacteriophages with the phage serial dilutions was performed each time in triplicate in 90 mm Petri dishes with the use of 3 independent bacterial cultures while using the same series of dilutions of the bacteriophage suspension. At the dropped spot, the possible observations are as follows: CL (clearance), T (turbidity), P (plaques), or NL (no clearance on the spot). Strain is classified as sensitive when the lowest phage titer at which clearance, turbidity, or plaques are observed is from 10^3^ to 10^5^ PFU/mL, intermediate when it is from 10^6^ to 10^8^ PFU/mL, and insensitive when no sign is observed up to 10^8^ PFU/mL.

### 2.7. The Effectiveness of Bacteriophage Cocktail in Prevention of Formation and Eradication of 24 h Bacterial Biofilm

The experiment was performed on the basic strain collection, including an additional 4 *E. coli* and 3 *S. aureus* strains resistant to one phage of cocktail components. The effectiveness of the phage cocktail in preventing the formation and eradication of bacterial biofilm was tested using a modified method described by Maszewska et al. [37]. Briefly, in the study of prevention of biofilm formation, 50 µL of 25-fold diluted 18–20 h bacterial cultures (2 × 10^8^ CFU/mL) and 50 µL of phage cocktail of titer 4 × 10^8^ PFU/mL were added simultaneously to a 96-well flat bottom polystyrene plate (MOI 50). Two identical plates were prepared in this manner. The first one was incubated for 24 h at 37 °C in a humidified chamber, and then the MTT test was performed as follows: The medium was removed from all wells, then each well was washed with 100 µL of 0.85% NaCl to rinse the planktonic forms of bacteria. Next, 100 µL of LB medium and 10 µL of 5 mg/mL MTT [3-(4,5-dimethyl-2-yl)-2,5 diphenyltetrazolium bromide] (MERCK, Burlington, MA, USA) were added to each well, and the plate was incubated for 30 min at 37 °C. Dehydrogenases in living cells convert yellow tetrazolium salts (MTT) to purple formazan crystals. After the incubation medium was discarded, 150 µL of DMSO (Chempur, Karlsruhe, Germany), 25 µL of 0.05 M glycine buffer pH 10.6 (0.05 M glycine (Chempur), and 0.043 M NaOH (Chempur) were added to wells to dissolve the formazan crystals. The absorbance value at 570 nm wavelength (OD_570_), which is directly proportional to the amount of viable cells adsorbed to the wells of the plate (formed biofilm), was measured. The second plate was used for evaluation of bacterial kinetic growth, and the optical density at 600 nm wavelength (OD_600_) was measured every 20 min for 24 h. In the 24 h biofilm eradication test, the plate was prepared similarly, but after the application of 100 µL of 50-fold diluted overnight bacterial cultures (2 × 10^8^ CFU/mL) to a 96-well plate, the plate was incubated for 24 h at 37 °C in a humidified chamber. Then, the suspensions were collected, and the biofilms were washed with sterile saline. Next, 100 µL of phage cocktail, titer 2 × 10^8^ PFU/mL, was added to appropriate wells, and the plate was again incubated for 24 h at 37 °C in a humidified chamber. Control wells contained 100 µL of the medium instead. After incubation, the MTT test was performed according to the same procedure. Each experiment was repeated three times. The calculations were made based on the following formula:$$E_{BC}=100\%-\left[{(A}_{TS}\times 100\%\right)\;:\;A_{C100\%}]$$
where:
$E_{BC}$—effectiveness of biofilm control,*A_TS_*—absorbance of test sample,*A*_*C*100%_—average absorbance of the 100% biofilm control.

### 2.8. The Effectiveness of Phage Cocktail in the Milk Environment

The study was carried out on store-bought milk (3.2% fat) pasteurized at low temperatures (Piatnica, Poland). A milk sample, about 300–400 mL, was centrifuged at 4500 rpm (swing-out rotor; 4754 RCF), at 4 °C, for 10 min. The obtained cream layer was removed, and the remaining milk was transferred into one sterile bottle. Milk fat percentage after centrifugation was not measured. Before the experiment, milk volume was portioned into 50 mL Falcon tubes, 18 mL each, and incubated at 37 °C, 140 rpm, for 10 min. Overnight incubated bacterial cultures were diluted in 0.85% NaCl to obtain 1 × 10^5^ CFU/mL and 1 × 10^4^ CFU/mL densities. An amount of 2 mL of each prepared bacterial suspension was added to separate tubes with milk (final titer 1 × 10^4^ CFU/mL and 1 × 10^3^ CFU/mL, respectively). The control sample contained 18 mL of milk and 2 mL of sterile 0.85% NaCl. After mixing, the samples were portioned in a volume of 9 mL into the two 50 mL Falcon tubes. Amounts of 1 mL of the phage cocktail (2 × 10^8^ PFU/mL) and 1 mL of sterile buffer solution (a control of bacterial growth in milk) were added to the appropriate tubes and incubated at 140 rpm, 37 °C, for 24 h. After incubation, the number of bacteria in milk samples was assessed using a selective MacConkey and Chapman medium. Four replicates were performed for each strain tested. To determine if there were significant differences between controls and each strain and tested initial density of bacteria, student T-tests were conducted, and *p* values < 0.05 were considered as significant, using the program Prism 10 version 10.0.3.

### 2.9. Storage Stability of the Bacteriophage Cocktail

The stability of the cocktail was tested at 2–10 °C for 24 months. The initial titer of the cocktail was 2 × 10^8^ PFU/mL. The measurements were made every 3 months, and the titer was determined by the double-layer agar method with one plate per dilution [19] as described in Section 2.2.

## 3. Results

### 3.1. Phage Characteristics

Bacteriophages covered in this article were exposed to bioinformatic analysis, which revealed that they perform a lytic cycle only and due to this are considered virulent (Table 3 and Table 4). The analysis allowed classification of phage 303Ecol101PP to Tequatrovirus teqdroes (NCBI txid: 2844259), 308Ecol101PP to Mosigvirus mar005p1 (NCBI txid: 2560437), and 351Saur083PP to Rosenblumvirus GRCS (NCBI txid: 2732598) species. For the rest of the analyzed phages, no direct references were detected, but their genera were identified: 310Ecol104PP is Tequatrovirus (NCBI txid: 10663), 348Ecol098PP is Mosigvirus (NCBI txid: 1913652), and 241Ecol014PP is Vequintavirus (NCBI txid: 1914852), while 355Saur083PP and 357Saur119PP are Kayvirus (NCBI txid: 1857843). The assemblies of genomes were deposited in the GenBank database (Table 3 and Table 4.). Genome maps and comparisons with reference genomes are available in Appendix A (https://doi.org/10.5281/zenodo.13772927).

Bioinformatic investigations were confirmed via transmission electron micrographs (TEM) visualization (Figure 1), which revealed the morphology of icosahedral capsid and a long contractile tail for all five anti-*E. coli* bacteriophages and two anti-*S. aureus* phages, and the icosahedral capsid and a short non-contractile tail for 351Saur083PP phage. These findings were consistent with in silico analyses. Next, the taxonomic similarity between isolated phages and their most similar reference was calculated via Virus Intergenomic Distance by VIRIDIC software v1.0, which is presented in Figure 2.

Further bioinformatic analyses allowed for the detection of the receptor-binding proteins for all analyzed phages, which in turn allow for the reconstruction of the phage’s receptor binding systems and the uncovering of the action of our phages in the first stage of infection (publication under preparation). In the case of phages attacking Gram-positive bacteria (351Saur083PP, 355Saur083PP, and 357Saur119PP), the analysis predicted affinity to the wall teichoic acids (WTA) decoration sugar (Table 5). The bioinformatic prediction suggests that Rosenblumvirus 351Saur083PP and Kayvirus 355Saur083PP recognize the same sugar motif but attach it by different proteins and mechanisms. Moreover, these phages use different supporting receptors to boost the attachment process. In the case of phage 357Saur119PP, which belongs to the same genus as 355Saur119PP, the bioinformatic studies detected crucial differences in receptor binding domains and suggested the recognition of other structural motifs (publication under preparation).

The analysis of phages specific to *E. coli* (241Ecol014PP, 303Ecol101PP, 308Ecol101PP, 310Ecol104PP, 348Ecol098PP) predicts that bacteriophages belonging to the same genus recognize the same second receptor (lipopolysaccharide sugar core), which is irreversibly binding by almost identical short tail fibers. However, their specificity is strictly related to the proteins, which create the final part of the long tail fibers that recognize external loops of the different outer membrane porins (Table 5).

### 3.2. Lytic Activity Results

The lytic activity of selected bacteriophages was assessed through the measurement of absorbance (OD_620_) of control samples (bacteria) compared to the absorbance of samples of bacteria treated with phage lysate on 96-well plates. The results are expressed as a percentage of bacterial collection with inhibited growth resulting from the treatment with a particular bacteriophage (Table 6). Among phages specific to *E. coli*, 348Ecol098PP showed the broadest inhibitory effect with strong inhibition concerning 50% of bacterial collection. The 241Ecol014PP, 308Ecoll01PP, and 310Ecol104PP bacteriophages significantly inhibited the growth of at least 30% of the *E. coli* strains, and the 303Ecol101PP phage strongly impeded the growth of 28% of tested bacterial strains. Noteworthy is the broad range of growth inhibition of the specific to *S. aureus* 355Saur083PP phage that covers over 85% of the tested bacterial collection. 351Saur083PP and 357Saur119PP caused strong inhibition of growth for 40% and 26.7% of tested strains, respectively. It is worth mentioning that the lytic activity of selected phages overlaps, providing a final coverage of the *E. coli* collection of 83%, and for *S. aureus*, bacterial collection of 100% (93.3% of strong inhibition and 6.7% of weak inhibition). Lytic activity data are available in Appendix A (https://zenodo.org/records/11047177; Appendix A: Endpoint OD difference for individual anti-*E. coli* bacteriophages of the cocktail; Appendix A: Endpoint OD difference for individual anti-*S. aureus* bacteriophages of the cocktail; Appendix A: Summary of lytic activity of tested bacteriophages towards the bacterial collection). For visualization of obtained results, analysis for two bacterial strains, *E. coli* 133 and *S. aureus* 083, was presented in Figure 3. These two strains were selected based on their sensitivity to tested phages.

### 3.3. Phage Resistance Induced by Selected Phages

The possibility of inducing resistance among bacteria was evaluated for tested bacteriophages. Host strains of each phage were exposed to an overdose of phages propagated on them. When inducing strain variants resistant to the tested phage, 20 *E. coli* and eight *S. aureus* strains insensitive to selected bacteriophages were obtained (Table 7). The spot test method performed on phage-resistant bacteria variants showed that among 20 analyzed *E. coli* strains, 14 remained sensitive towards at least two other bacteriophages, and five strains were sensitive to one bacteriophage. Interestingly, the 357Saur119PP phage was observed not to cause phage resistance in all analyzed *S. aureus* strains and retained its lytic activity towards strains that are resistant to two others specific to *S. aureus* phages. The obtained results are summarized in Table 7.

To discover the phages-resistance mechanism induced by selected phages, the obtained resistant variants were exposed to sequencing together with wild type strains. For bioinformatic analysis, only three *S. aureus* (Table 8) and four *E. coli* (Table 9) phage-resistant mutant’s NGS data were used. The genomes’ sequences of wild type strains and NGS data of mutants are available in NCBI BioProject ID: PRJNA1162721. Other sequences were withdrawn from analysis due to the low quality of sequences or inconsistent observations. Regarding bacterial strains resistant to phage 351Saur083PP and 355Saur083PP (*S. aureus* 119PP2018, 120PP2018, and 121PP2018), analysis revealed a mutation in the region encoding poly(ribitol-phosphate) β-N-acetylglucosaminyltransferase (TarS), which is an enzyme attaching the N-acetylglycosamine branch attached by β 1–4 bound. Since mutants remained sensitive to phage 357Saur119PP it can be implied that this phage binds to another sugar motif (Table 8). Additionally, some mutations were identified in the sequence encoding partial 5S ribosomal RNAs; however, they have no influence on bacteria-phage interaction so were ignored in this study.

The analysis performed on strain *E. coli* 235PP2017, which is resistant to phage 308Ecol101PP, revealed a deletion in the region encoding outer membrane porin C. This mutation causes the frameshift and induces premature translation termination. The resistance of this mutant suggests that the upper phage recognizes OmpC as a main receptor. The strains *E. coli* 1304PP2022 and 1306PP2022, which are generated by exposition to phage 303Ecol101PP, have a mutation in the region encoding regulator of the length of the O-antigen component of lipopolysaccharide chains and knockout mutation in the ompC gene. The bioinformatic studies of phage 303Ecol101PP show that it does not recognize O-antigen but is strictly dependent on OmpC protein, which should be its main receptor. The analysis of phage-resistant mutants confirmed this prediction. The bioinformatic analysis of *E. coli* 265PP2018 detected mutation in two genes. The first is the duplication of four nucleotides in the region encoding the ompA gene, which causes a frameshift and induces an appearance of stop codon and premature translation termination. The second detected mutation is the single nucleotide deletion in the region encoding UDP-Gal:alpha-D-GlcNAc-diphosphoundecaprenol beta-1,3-galactosyltransferase. This enzyme catalyzes the addition of galactose, the second sugar moiety of the O7-antigen repeating unit, to GlcNAc-pyrophosphate-undecaprenol. According to bioinformatic analysis, *E. coli* 265PP2018 strain (resistant to phage 310Ecol104PP) was predicted to recognize OmpA as a receptor, which was confirmed by NGS data analysis. The results are summarized in Table 9.

### 3.4. Phage Cocktail Host Range

Based on the in-depth bioinformatic characterization of bacteriophages and analysis of their lytic activity and influence on bacteria resistance occurring, it was inferred that they are solid candidates for cocktail components. As a consequence, the cocktail composed of them was tested for specificity to bacteria from the bacterial collection described in the paragraph on bacterial collection and growth conditions of the Methods section. A cocktail specificity test was carried out via a serial dilutions spot test. Among tested *E. coli* strains, two were considered insensitive (*E. coli* 091, *E. coli* 096), two were of medium sensitivity (*E. coli* 099, *E. coli* 118), and 13 remained sensitive. In the case of *S. aureus*, all 15 tested strains were considered sensitive. The summary of the results expressed as a percentage of the number of strains is presented in Figure 4.

### 3.5. The Effectiveness of Bacteriophage Cocktail in Preventing Biofilm Formation and Eradicating 24 h Bacterial Biofilm

A developed anti-mastitis cocktail was used to evaluate its biofilm degradation ability. The experiment showed that within *E. coli* strains, inhibition of biofilm formation of a minimum of 50% was observed for 50% of analyzed strains (Figure 5A). The bacteriophage cocktail also destroyed a minimum 50% of 24 h biofilm for 59% of *E. coli* strains (13 out of 22). Regarding *S. aureus* strains, the test demonstrated almost 100% inhibition of biofilm formation for all tested strains and bacterial biofilm eradication of 50% in the case of 39% of tested strains (7 out of 18) (Figure 5B).

In parallel with the prevention of the biofilm formation test, the bacterial kinetic growth was measured (OD_600_). The growth curves for *E. coli* 133 and *S. aureus* 083 strains treated with an anti-mastitis bacteriophage cocktail are shown in Figure 5C,D, and results for all tested bacterial strains are included in Appendix A (https://doi.org/10.5281/zenodo.11047238; Appendix A). Results obtained during 24 h measurement showed that the phage cocktail inhibited the growth of 18 out of 18 *S. aureus* tested strains and four out of 22 *E. coli* strains. For six *E. coli* strains, the phage cocktail caused partial inhibition of bacterial growth, and for four other *E. coli* strains, it caused growth inhibition during the first hours of incubation, and then the kinetics of bacterial growth were identical to those of a strain not treated with the cocktail. In four out of 40 tested strains, no effect of the anti-mastitis phage cocktail on bacterial growth kinetics was observed.

### 3.6. Cocktail Effectiveness in Bacteria Eradication in the Milk Environment

In this study, two representative strains (*E. coli* 133 and *S. aureus* 083) were used. The effectiveness of the bacteriophage cocktail in bacteria eradication in the milk environment was defined as % of log of bacteria number reduction compared to the log number of bacteria in the control sample assumed as 100%. Two different initial bacterial densities were tested (1 × 10^3^ CFU/mL and 1 × 10^4^ CFU/mL). Differences in Log_10_(CFU/mL) for each strain and initial bacterial density tested compared to control samples with SM buffer are presented in Figure 6A. The applied phage cocktail was more effective in reducing the number of *S. aureus* 083 reaching 43% reduction (*p* < 0.001 for 1 × 10^3^ CFU/mL and *p* < 0.01 for 1 × 10^4^ CFU/mL), while for the *E. coli* strain, the reduction ranged approximately 30% (*p* < 0.001 for 1 × 10^3^ CFU/mL and *p* < 0.05 for 1 × 10^4^ CFU/m). In control samples, after 24 h incubation, bacterial growth was observed in the range of 4 × 10^7^ CFU/mL to 2 × 10^8^ CFU/mL for *E. coli* and 3 × 10^8^ CFU/mL to 3 × 10^9^ CFU/mL for *S. aureus*. The initial bacterial density values had no impact on the effectiveness of the cocktail used (Figure 6B). In addition, the difference in the appearance of milk samples artificially infected with *S. aureus* between treated or not treated with the phage cocktail was observed (Figure 6C). Precipitation of milk proteins was observed in samples without phages.

### 3.7. Storage Stability of Bacteriophage Cocktail

The stability study was carried out for 24 months at 2–10 °C with measurements taken every 3 months. It was expressed as a percentage of the logarithm of the PFU/mL for each measurement time point in reference to the initial titer. After 24 months, stability at a level of at least 94% was observed (Table 10) with the stability of the anti-*E. coli* component being at a higher level than the anti-*S. aureus* one.

## 4. Discussion

*E. coli* and *S. aureus* are among the most frequently mentioned mastitis-causing bacterial pathogens [38,39]. *E. coli* is the most common Gram-negative pathogen responsible for acute clinical mastitis in dairy cows during early lactation and some subclinical phenotypes, whereas *S. aureus* is of great concern because of its contagiousness, persistence in the environment, high colonization abilities, and generally poor response to therapies [2,5].

This study aimed at developing a phage cocktail directed against bacteria associated with bovine mastitis, especially *E. coli* and *S. aureus*. Eight isolated phages (five specific to *E. coli* and three specific to *S. aureus*), originating from environmental samples of wastewater, underwent in-depth bioinformatic characterization and in vitro evaluation.

In silico analysis of selected phages confirmed their virulent nature, while analysis of phages and spontaneously generated phage-resistant mutants derived from host bacteria allowed for the determination of the main phage receptors, what is summarized in Table 11 (compilation of Table 5, Table 8 and Table 9, from the Results section). The predictions were highly precise for five phages (303Ecol101PP, 310Ecol104PP, 308Ecol101PP, 351Saur083PP, and 355Saur083PP). In the case of 348Ecol098PP, the prediction was correct for the irreversible binding receptor (LPS core), which is bound by the short tail fibers, but unprecise for the reversible binding receptor, recognized by the long tail fibers. For two phages (241Ecol014PP and 357Saur119PP), the lack of phage-resistant mutant sequences did not allow for precise phage receptor prediction. In the case of 241Ecol014PP, the situation was additionally complex due to complicated architecture of the base plate and tail fiber and the low quality of available structural data of evolutionary closely related phages.

Combining phages that recognize different conservative structural elements into a cocktail allows for the recognition of a wide range of targeted bacteria. What is worth underlining is that these conservative structural elements also play the role of virulence factors (OmpA, OmpC, LPS, and WTA), so combing phages that recognize these structures focuses forces on highly virulent pathogens. In addition, as was detected in many studies, if resistance mechanisms rely on knocking out these genes, even if phage resistance occurs, the mutants are less virulent [40].

This approach for phage cocktail designing is the new direction. Interestingly, Dinesh Subedi and colleagues [41], who developed enhanced phage cocktail using phage training and expanded host range through targeted phage isolation against low-coverage strains, also focused on the detection of phages’ receptors and their in-depth analysis. Our and Subedi’s results confirm that this approach is efficient for designing effective phage cocktails that escape the phage resistance mechanism in bacteria. Such a combination of phages into a cocktail allows for obtaining the maximum spectrum of action, which justifies the scheme proposed in Figure 7 describing the synergistic effect of appropriately selected phages.

The phage cocktail composed in such a way triggers a highly targeted effect, which was proven on a well-characterized collection of bacterial strains isolated from livestock with mastitis symptoms [18]. Our results showed that the developed anti-mastitis cocktail has specificity against tested *S. aureus* strains reaching 100% and almost 86% in the case of *E. coli* strains, if the values for sensitive and medium sensitive strains are combined (12 out of 14 strains). Another study with bacteriophages specific for *Staphylococcus* spp. isolated from dairy cattle also demonstrated their broad spectrum of antibacterial activity and indicated them as potential tools in maintaining environmental homeostasis [42]. This seems to imply that field phages with a wide range of hosts are promising candidates for the development of anti-mastitis therapies, and the suggested approach is consistent with this paper.

Titze et al. [6] suggested that a possible reason for incomplete host coverage might be insufficient differentiation of phages in terms of their origin or family membership, and optimal composition of the phage cocktail should include representatives of all families. This suggestion is partially reflected in the case of our phage cocktail, as specific to *E. coli* phages represent three genera within the *Straboviridae* family, and anti-*S. aureus* phages represent the *Rountreeviridae* family and two genera of the *Herelleviridae* family.

Further, the bactericidal activity of the phage cocktail was evaluated with in vitro tests. A common obstacle in antibacterial therapies is the ability of bacteria to form biofilms, which impede eradication by protecting themselves from biological, chemical, and physical factors and by increasing resistance to antimicrobial agents and the host immune response. Both the *E. coli* and *S. aureus* are capable of biofilm formation [43]. In the current study, the effectiveness of the developed bacteriophage cocktail in preventing biofilm formation and eradicating 24 h bacterial biofilm was evaluated. Concerning *E. coli* strains, the study demonstrated inhibition of biofilm formation of a minimum of 50% for 50% of analyzed strains and destruction of a minimum of 20% of 24 h biofilm for 86% of tested strains and a minimum of 50% of biofilm for 59% of *E. coli* strains. Phages specific to *S. aureus* demonstrated 99% inhibition of biofilm formation for all tested strains and biofilm eradication at a minimum level of 20% for 67% of *S. aureus* strains and eradication of 50% in the case of 39% of tested strains. The obtained results are consistent with the current literature providing data confirming the potential of bacteriophages in biofilm eradication [44,45,46]. In addition, numerous reports suggest that bacteriophage-derived depolymerases play an important role against biofilms due to their ability to recognize, bind, and degrade the polysaccharide compounds of bacterial cell walls, therefore improving phage penetration [47,48,49]. A characteristic feature of depolymerase-producing phages is halos surrounding the plaques, caused by the diffusion of the enzyme into the medium and depriving the bacteria of the envelope [50]. All phages specific to *S. aureus* included in the cocktail possess genes coding polysaccharide depolymerases (confirmed by genetic analysis) specific to different sugar motifs. However, only 351Saur083PP phage forms plaques surrounded by a halo on the bacterial lawn, which suggests its ability to produce active depolymerases (Figure 8).

There are continuously emerging studies highlighting the therapeutic potential of bacteriophages for mastitis-related bacterial strains, including those resistant to commonly used antibiotics [51], assessed by in vitro and in vivo experiments [52,53,54]. A study with a cow with mastitis caused by *E. coli* resistant to drugs demonstrated that a cocktail of three phages greatly reduced the number of bacteria itself but also limited somatic cell counts and inflammatory factors and alleviated the symptoms of mastitis in cow [2].

The annual usage of antibiotics that are the first line of treating mastitis in dairy cattle can vary significantly depending on the region, farm practices, and the prevalence of the disease. The specific global or national figures are hardly readily available. However, it has been estimated that mastitis treatment constitutes a major part of the total antibiotic usage in the dairy industry in Europe as well as in the United States [55,56]. Several key factors must be taken into account when considering the expected reduction in antibiotic usage, including the effectiveness of the phage therapy, the specific bacterial strains involved, and the extent to which phage therapy would be adopted by healthcare providers on farms.

It was shown that the therapeutic efficacy of a phage cocktail can be comparable to that of the antibiotic ceftiofur sodium for *S. aureus*-induced mastitis in mice and *E. coli*-¬associated mastitis in cattle [2,57]. Moreover, lytic phages were demonstrated to resuscitate an ineffective antibiotic for previously resistant bacteria while simultaneously synergizing with antibiotics in a class-dependent manner [58].

All in all, the estimation of antibiotic usage reduction must be taken carefully since it is still based on ongoing research, and the actual reduction will eventually depend on the specific context. While exact figures can vary, the consensus in the scientific community is that phage therapy holds great promise for reducing antibiotic usage in the treatment of mastitis and other bacterial infections [59].

On the other hand, there are reports vaguely confirming phage activity in milk [60,61]. Nale and McEwan [62] mention that clumping of *S. aureus* on fat globules within the milk may somehow provide them with a protective barrier against phage attachment. They also indicate that whey proteins in milk can adhere to the surface of *S. aureus* cells, therefore blocking the potential receptors from attachment of phages. Thus, it was important to evaluate the effectiveness of an anti-mastitis phage cocktail in the research setup that imitates the natural environment for phages. In the case of chronic mastitis, *S. aureus* is excreted into the milk during milking. In properly drawn milk, the bacterial counts range from 100 to 200 CFU/mL; however, in the case of infected udders, these counts may increase significantly, reaching up to 10^4^ CFU/mL or in extreme cases up to 10^8^ CFU/mL [63]. In addition, Peles et al. [64] indicate *S. aureus* in bulk tank milk at levels of up to 10^3^ CFU/mL. Since our phage cocktail could be used in the future during the dry period, we decided to test the setup with two different initial bacterial densities (1 × 10^3^ CFU/mL and 1 × 10^4^ CFU/mL). Our current phage cocktail was demonstrated to retain its antibacterial activity in a cow’s milk environment, reducing the number of bacteria on average by up to 43% for *S. aureus* and up to 30% for *E. coli*. Titze [6] achieved similar reduction values in a pasteurized milk study with a mixture of three lytic anti-*S. aureus* phages. There were no statistically important differences between tested initial bacterial densities. Additionally, the presence of bacteriophages prevented the precipitation of milk proteins and visually improved the quality of milk samples. These insights justify the continued evaluation of the effectiveness of the developed phage cocktail with in vitro models, including different MOI values to reflect various environmental situations.

Since productive phage infection, i.e., reaching the highest possible titer of phages can be achieved if a sufficient number of bacteriophages reach the host bacteria, phages should be administered directly to the site of infection. Intravenous administration of phages will allow only a small fraction of them to reach the target site of infection because they are quickly removed from the bloodstream and demonstrate a slow diffusion rate [65]. In the case of mastitis, the highest concentration of bacteria is in the infected udder; therefore, the intramammary route of administration seems to be the most reasonable. This is in line with the study of staphylococcal mastitis in ewes, where the comparison of the efficacy of intramammary and intramuscular antibiotic injection revealed some benefit for the former [66]. The phage cocktail presented in this study is likely to be administered intramammarily, yet the decision on its final formulation has not yet been made. In addition, it seems possible that the higher volume of the cocktail, not particularly the higher titer of phages, contributes to better distribution in vitro and ultimately helps bacteriophages to track the bacteria host cells [6]. This issue should be taken into account when designing the in vivo experiments, bearing in mind that the volume of the preparation should be evenly distributed within the udder. In vivo studies are planned to be conducted as a next stage of development phase, as different aspects of developed product need to be checked in the natural environment. First of all, it is crucial to test its safety, tolerance, and then the effectiveness with selected therapeutic dose, volume, and delivery system. Currently, there are no effective preventatives of mammary bacterial infections available. Despite years of research, no effective vaccines have been developed [67]. Besides antibiotic therapy, which remains the most common method, multiple approaches to manage and prevent mastitis infections have been applied [68,69,70]. Unfortunately, neither is satisfactory when applied as a monotherapy, implicating the need for combination therapies in the future. Perhaps this is the gap that can be filled with phage therapy, as it has already been suggested that the synergistic effect of phage and antibiotic therapy may lead to a reduction of bacterial resistance to phages and antibiotics [6]. However, the clue is the appropriate composition of the phage cocktail.

## 5. Conclusions

There is currently a shift away from antibiotic therapies in animal husbandry, which opens up opportunities for alternative trends in animal treatment. Phage therapy seems to be attracting more and more attention due to the high specificity towards the pathogen and the growing number of promising in vitro and in vivo studies. The approach for phage cocktail design proposed in this study seems to be crucial for the development of highly effective solutions for bacteria pathogens. In silico assessment of bacteriophage genomes and phage-host interactions allows for the appropriate selection of phages for the cocktail and constitutes a new direction in the procedure of phage cocktail design, which, supported by microbiological analyses, provides the foundation for effective phage products.

The developed anti-mastitis phage cocktail was composed in such a way as to obtain the highest possible efficacy and to avoid the emergence of bacteria resistance, which is the most common obstacle in the case of antibiotic therapies. The results obtained from in vitro studies showing a wide host range and strong lytic activity against a well-characterized bacterial collection indicate the therapeutic potential of the developed cocktail in the treatment of bovine mastitis that could limit the impact of bacterial disease on animal and human health.

## Figures and Tables

**Figure 1 pathogens-13-00839-f001:**
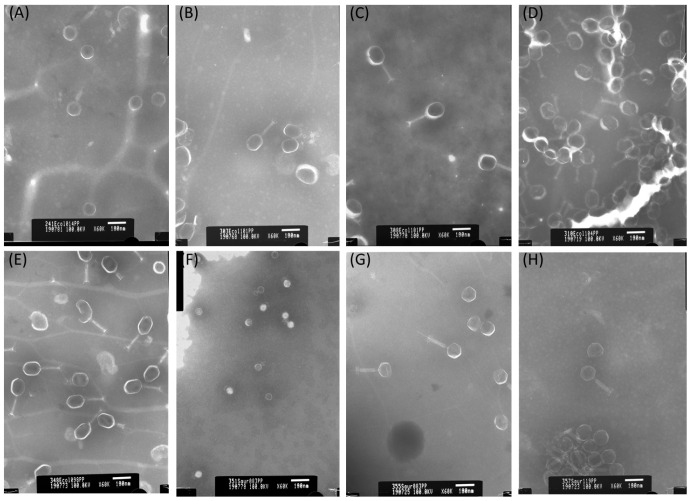
TEM micrographs (magnification 60,000×) of isolate bacteriophages: (**A**) 241Ecol014PP, (**B**) 303Ecol101PP, (**C**) 308Ecol101PP, (**D**) 310Ecol104PP, (**E**) 348Ecol098PP, (**F**) 351Saur083PP, (**G**) 355Saur083PP, (**H**) 357Saur119PP.

**Figure 2 pathogens-13-00839-f002:**
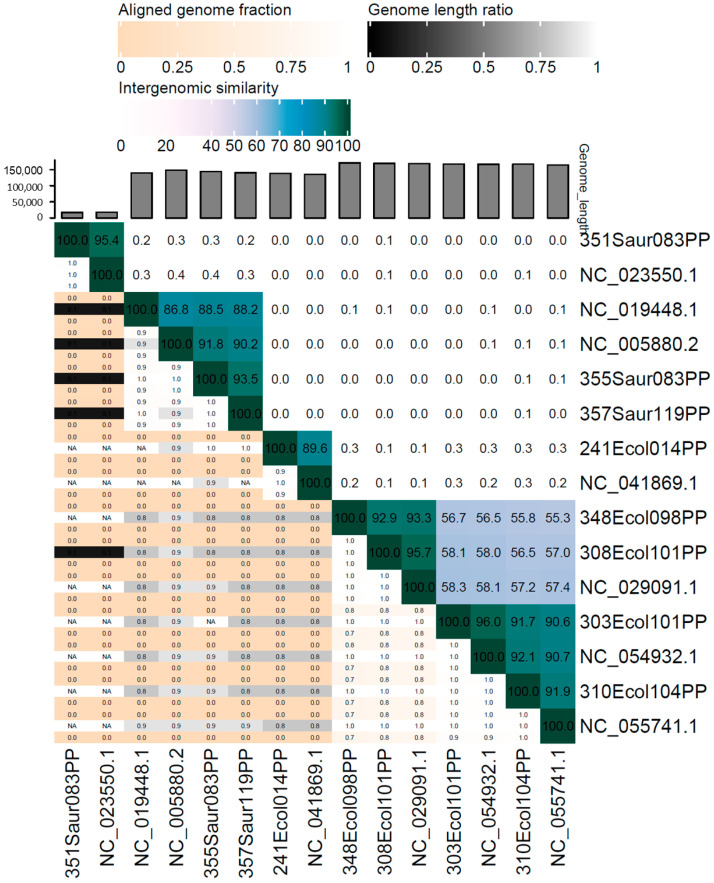
The results of Virus Intergenomic Distance calculation between analyzed phages and the most similar reference bacteriophages.

**Figure 3 pathogens-13-00839-f003:**
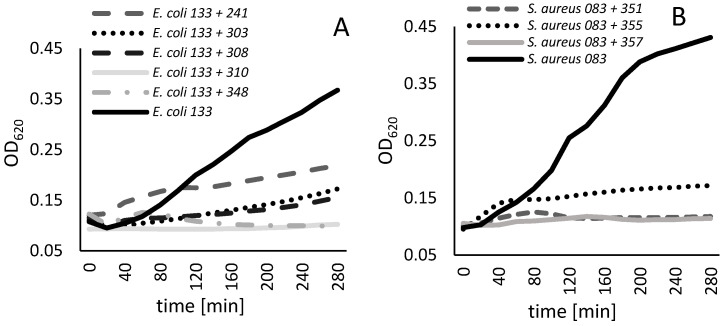
Growth curves of *E. coli* 133 (**A**) and *S. aureus* 083 (**B**) cultures treated with selected bacteriophages.

**Figure 4 pathogens-13-00839-f004:**
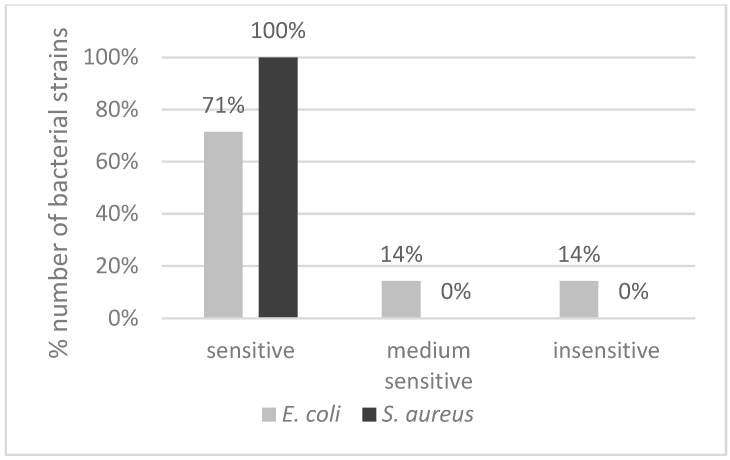
The sensitivity of *E. coli* and *S. aureus* bacterial strains to anti-mastitis bacteriophage cocktail presented as % number of strains that are sensitive, medium sensitive, or insensitive among all tested strains.

**Figure 5 pathogens-13-00839-f005:**
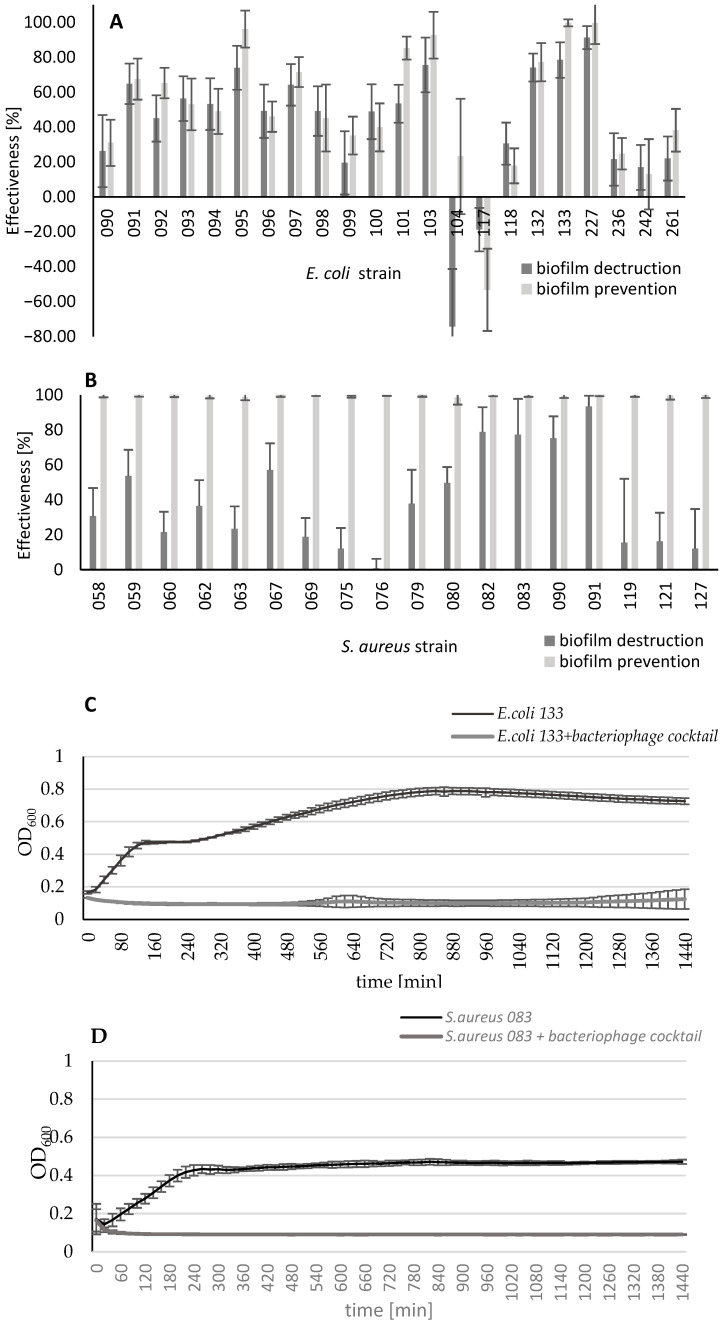
The effectiveness of bacteriophage cocktail in biofilm prevention and eradication for *E. coli* (**A**) and *S. aureus* (**B**) strains. The growth curves of *E. coli* 133 (**C**) and *S. aureus* 083 (**D**) cultures with an anti-mastitis phage cocktail.

**Figure 6 pathogens-13-00839-f006:**
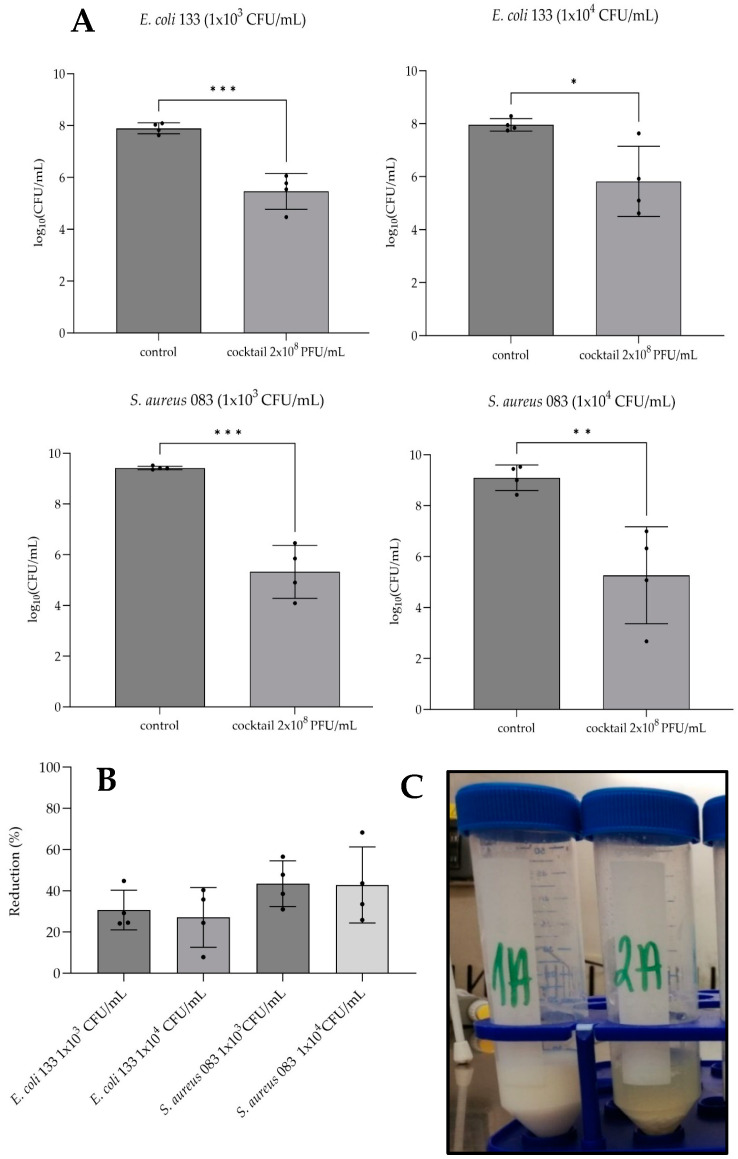
The effectiveness of anti-mastitis cocktail on *E. coli* 133 and *S. aureus* 083 bacterial strains in the milk model study. (**A**) Comparison of log_10_(CFU/mL) of tested variants to control samples without bacteriophage cocktail. Statistical differences are displayed on the graph (* *p* < 0.05; ** *p* < 0.01; *** *p* < 0.001). (**B**) Reduction of bacterial growth expressed as %Log_10_. Data presented are average results from four repeats (gray bars); individual results are shown as black dots, and error bars represent the standard error of the mean. (**C**) The appearance of milk samples infected with *S. aureus* 083 with (1A) and without (2A) phage cocktail.

**Figure 7 pathogens-13-00839-f007:**
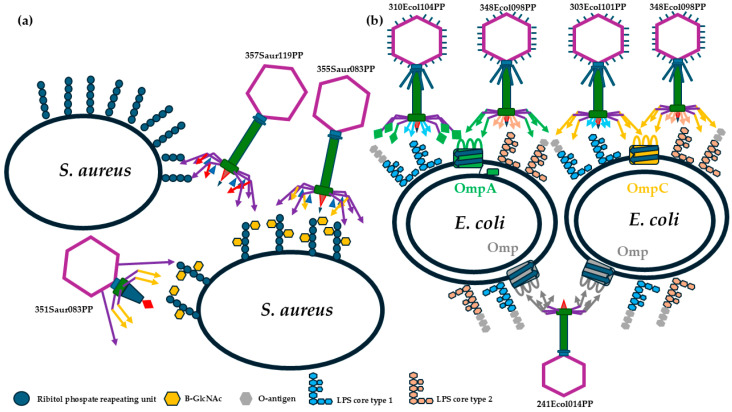
The graphical representation of the synergistic effect of bacteriophage components in the designed anti-mastitis cocktail. (**a**) Gram-positive phages 355Saur083PP and 351Saur083PP are dependent on the same sugar branches of WTA but expose different supporting proteins and depolymerase for recognition and digestion of the biofilm. Phage 357Saur119PP recognizes the most conservative receptor and is the core of the anti-Gram positive cocktail. (**b**) The bacteriophages against Gram-negative bacteria were selected for recognition of main outer membrane porins in *E. coli* and two types of LPS core. The phage 241Ecol014PP plays a similar role as 357Saur119PP and recognizes *E. coli* as resistant to the rest of the phages against *E. coli*.

**Figure 8 pathogens-13-00839-f008:**
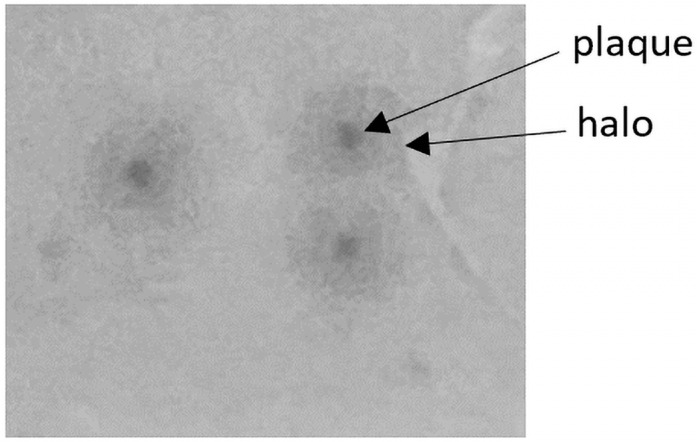
Plaques formed by 351Saur083PP bacteriophage.

**Table 1 pathogens-13-00839-t001:** Bacterial collection.

Bacterial Strain	UWM Strain ID	Year of Isolation
*Escherichia coli* 090PP2016	407	2016
*Escherichia coli* 091PP2016	408	2016
*Escherichia coli* 092PP2016	411	2016
*Escherichia coli* 093PP2016	412	2016
*Escherichia coli* 094PP2016	384	2016
*Escherichia coli* 095PP2016	230	2016
*Escherichia coli* 096PP2016	375	2016
*Escherichia coli* 097PP2016	381	2016
*Escherichia coli* 098PP2016	125	2016
*Escherichia coli* 099PP2016	513	2016
*Escherichia coli* 100PP2016	282	2016
*Escherichia coli* 101PP2016	124	2016
*Escherichia coli* 103PP2016	419	2016
*Escherichia coli* 104PP2016	418	2016
*Escherichia coli* 117PP2016	552	2016
*Escherichia coli* 118PP2016	551	2016
*Escherichia coli* 132PP2017	538	2017
*Escherichia coli* 133PP2017	574	2017
*Staphylococcus aureus* 058PP2016	377	2016
*Staphylococcus aureus* 059PP2016	360	2016
*Staphylococcus aureus* 060PP2016	342	2016
*Staphylococcus aureus* 062PP2016	312	2016
*Staphylococcus aureus* 063PP2016	322	2016
*Staphylococcus aureus* 067PP2016	390	2016
*Staphylococcus aureus* 069PP2016	522	2016
*Staphylococcus aureus* 075PP2016	476	2016
*Staphylococcus aureus* 076PP2016	227	2016
*Staphylococcus aureus* 079PP2016	294	2016
*Staphylococcus aureus* 080PP2016	165	2016
*Staphylococcus aureus* 082PP2016	228	2016
*Staphylococcus aureus* 083PP2016	536	2016
*Staphylococcus aureus* 090PP2016	544	2016
*Staphylococcus aureus* 091PP2016	556	2016

**Table 2 pathogens-13-00839-t002:** Bacteriophages used in the study.

Bacteriophage	Short Name	Host	Source of Sample
241Ecol014PP	241	*E. coli*	Water vacuum cleaner
303Ecol101PP	303		Waste water ^1^
308Ecol098PP	308		Waste water ^2^
310Ecol104PP	310		Waste water ^1^
348Ecol098PP	348		Waste water ^1^
351Saur083PP	351	*S. aureus*	Waste water ^2^
355Saur083PP	355		Waste water ^2^
357Saur119PP	357		Waste water ^2^

Source of sample: ^1^ sewage treatment plant—Department of Municipal and Housing Management in Stryków, Poland, ^2^ wastewater treatment plant—GOS in Łódź, Poland.

**Table 3 pathogens-13-00839-t003:** Genetic characterization of bacteriophages specific to *E. coli*.

Feature (GenBank: ID)		303Ecol101PP (OR062944)	308Ecol101PP (OR062945)	310Ecol104PP (OR062946)	348Ecol098PP (OR062947)	241Ecol014PP (OR062943)
Size of genome [bp]		166,904	169,543	167,023	170,844	138,401
ORF		265	270	258	265	208
tRNA		11	2	10	2	4
GC pairs content [%]		35	38	36	38	44
Taxonomy	*Class*	*Caudoviricetes*	*Caudoviricetes*	*Caudoviricetes*	*Caudoviricetes*	*Caudoviricetes*
	*Family*	*Straboviridae*	*Straboviridae*	*Straboviridae*	*Straboviridae*	*-*
	*Subfamily*	*Tevenvirinae*	*Tevenvirinae*	*Tevenvirinae*	*Tevenvirinae*	*Vequintavirinae*
	*Genus*	*Tequatrovirus*	*Mosigvirus*	*Tequatrovirus*	*Mosigvirus*	*Vequintavirus*
	*Species*	*Tequatrovirus teqdroes*	*Mosigvirus mar005p1*	*-*	*-*	*-*
NCBI Reference Sequence:		NC_054932.1	NC_029091.1	NC_054932.1	NC_029091.1	NC_041869.1
VIRIDIC [%]		95.97	95.67	92.07	93.27	89.62

**Table 4 pathogens-13-00839-t004:** Genetic characterization of bacteriophages specific to *S. aureus*.

Feature (GenBank: ID)		351Saur083PP (OR062948)	355Saur083PP (OR062949)	357Saur119PP (OR062950)
Size of genome [bp]		17,209	143,709	140,580
ORF		19	216	209
tRNA		0	4	4
GC pairs content [%]		29	30	30
Taxonomy	*Class*	*Caudoviricetes*	*Caudoviricetes*	*Caudoviricetes*
	*Family*	*Rountreeviridae*	*Herelleviridae*	*Herelleviridae*
	*Subfamily*	*Rakietenvirinae*	*Twortvirinae*	*Twortvirinae*
	*Genus*	*Rosenblumvirus*	*Kayvirus*	*Kayvirus*
	*Species*	*Rosenblumvirus GRCS*	*-*	*-*
NCBI Reference Sequence:		NC_023550.1	NC_005880.2	NC_005880.2
VIRIDIC [%]		95.40	91.83	90.22

**Table 5 pathogens-13-00839-t005:** The results of detection of receptor bindings elements and phage receptors prediction by protein sequence and models similarity.

Bacteriophages	Genus	Receptor Binding Proteins	Predicted Receptor
351Saur083PP	*Rosenblumvirus*	WLY86749.1; WLY86757.1; WLY86759.1; WLY86760.1; WLY86762.1	Main: β-1,4-GlcNAc and cell membrane; Support: unknown oligosaccharide
355Saur083PP	*Kayvirus*	WLY86864.1; WLY86866.1; WLY86868.1; WLY86875.1	Main: β-1,4-GlcNAc and cell membrane; Support: unknown oligosaccharide
357Saur119PP	*Kayvirus*	WLY87089.1; WLY87091.1; WLY87093.1; WLY87100.1	Main: α-1,4-GlcNAc and cell membrane; Support: unknown oligosaccharide
303Ecol101PP	*Tequatrovirus*	WLY85731.1; WLY85819.1; WLY85826.1; WLY85828.1	Main: lipopolysaccharide core the same as phage 310Ecol104PP and OmpC
308Ecol101PP	*Mosigvirus*	WLY86025.1; WLY86032.1; WLY86034.1; WLY86213.1	Main: lipopolysaccharide core the same as phage 348Ecol098PP and OmpC
310Ecol104PP	*Tequatrovirus*	WLY86436.1; WLY86435.1; WLY86348.1; WLY86341.1; WLY86339.1	Main: lipopolysaccharide core the same as 303Ecol101PP and OmpA and PhoE
348Ecol098PP	*Mosigvirus*	WLY86505.1; WLY86512.1; WLY86514.1; WLY86683.1	Main: lipopolysaccharide core the same as phage 308Ecol101PP and OmpC or OmpF or maltoporin
241Ecol014PP	*Vequintavirus*	WLY85594.1; WLY85596.1; WLY85598.1; WLY85602.1; WLY85606.1; WLY85608.1; WLY85610.1	Lipopolysaccharide and unknown outer membrane porin

**Table 6 pathogens-13-00839-t006:** Summary of lytic activity of tested bacteriophages towards the bacterial collection.

% of Strains within the Bacterial Collection	% of Strains within the Bacterial Collection		
Bacteriophage	Strong Inhibition	Weak Inhibition	No Influence
241Ecol014PP	33.3%	22.2%	44.4%
303Ecol101PP	27.8%	22.2%	50.0%
308Ecol101PP	44.4%	5.6%	50.0%
310Ecol104PP	44.4%	11.1%	44.4%
348Ecol098PP	50.0%	5.6%	44.4%
Overlapping activity of all *E. coli* phages	83.3%	0%	16.7%
351Saur083PP	40.0%	20.0%	40.0%
355Saur083PP	86.7%	6.7%	6.7%
357Saur119PP	26.7%	13.3%	60.0%
Overlapping activity of all *S. aureus* phages	93.3%	6.7%	0%

**Table 7 pathogens-13-00839-t007:** Sensitivity of resistant bacterial variants to tested bacteriophages. Explanation of the symbols in the table: + sensitive; − insensitive; ⊗ not tested.

Bacterial Strain Used to Obtain Phage Resistant Variants	Bacteriophage	Obtained Resistant Variant to Specific Bacteriophage	Phage Used in Spot Test							
			303	308	310	348	241	351	355	357
*E. coli* 101	303	*E. coli* 227PP2017	−	+	+	+	−	⊗	⊗	⊗
		*E. coli* 228PP2017	−	+	+	+	−	⊗	⊗	⊗
		*E. coli* 229PP2017	−	+	+	+	−	⊗	⊗	⊗
*E. coli* 095	303	*E. coli* 230PP2017	−	−	+	+	+	⊗	⊗	⊗
		*E. coli* 1304PP2022	−	−	+	+	+	⊗	⊗	⊗
		*E. coli* 1306PP2022	−	−	+	+	+	⊗	⊗	⊗
*E. coli* 095	308	*E. coli* 235PP2017	+	−	+	+	+	⊗	⊗	⊗
		*E. coli* 236PP2017	+	−	+	+	+	⊗	⊗	⊗
		*E. coli* 237 PP2017	+	−	+	+	+	⊗	⊗	⊗
*E. coli* 098	348	*E. coli* 241PP2017	−	+	+	−	+	⊗	⊗	⊗
		*E. coli* 242PP2017	−	−	−	−	+	⊗	⊗	⊗
		*E. coli* 243PP2017	−	−	−	−	+	⊗	⊗	⊗
		*E. coli* 244PP2017	−	−	−	−	+	⊗	⊗	⊗
*E. coli* 242	241	*E. coli* 261PP2018	−	+	−	+	−	⊗	⊗	⊗
		*E. coli* 262PP2018	−	+	−	−	−	⊗	⊗	⊗
		*E. coli* 263PP2018	−	+	−	−	−	⊗	⊗	⊗
		*E. coli* 264PP2018	−	−	−	−	−	⊗	⊗	⊗
*E. coli* 104	310	*E. coli* 265PP2018	−	+	−	−	+	⊗	⊗	⊗
		*E. coli* 266PP2018	−	+	−	−	+	⊗	⊗	⊗
		*E. coli* 300PP2018	−	+	−	−	+	⊗	⊗	⊗
*S. aureus* 083	351	*S. aureus* 119PP2018	⊗	⊗	⊗	⊗	⊗	−	−	+
		*S. aureus* 120PP2018	⊗	⊗	⊗	⊗	⊗	−	−	+
		*S. aureus* 121PP2018	⊗	⊗	⊗	⊗	⊗	−	−	+
		*S. aureus* 122PP2018	⊗	⊗	⊗	⊗	⊗	−	−	+
	355	*S. aureus* 123PP2018	⊗	⊗	⊗	⊗	⊗	−	−	+
		*S. aureus* 124PP2018	⊗	⊗	⊗	⊗	⊗	−	−	+
		*S. aureus* 125PP2018	⊗	⊗	⊗	⊗	⊗	−	−	+
		*S. aureus* 126PP2018	⊗	⊗	⊗	⊗	⊗	−	−	+
	357	resistant variants not obtained	⊗	⊗	⊗	⊗	⊗	⊗	⊗	⊗

**Table 8 pathogens-13-00839-t008:** SNP (single nucleotide polymorphism) detection in mutants derived from *S. aureus* 083PP2018. NT_POS—position in the mutated gene. AA_POS—position of changed amino acids. The effect of the mutation: *—stop codon.

Bacterial Mutants	NT_POS	AA_POS	Mutation Effect	Gene	Product
119PP2018	1177/1722	393/573	Stop gained Arg393 *	*tar*S	Poly(ribitol-phosphate) β-N-acetylglucosaminyltransferase TarS
	-	-	Intergenic region C1550T	5SrRNA	5S ribosomal RNA (partial)
120PP2018	1027/1722	343/573	Stop gained G1027T Glu343 *	*tar*S	Poly(ribitol-phosphate) β-N-acetylglucosaminyltransferase TarS
	-	-	Intergenic region C1550T	5SrRNA	5S ribosomal RNA (partial)
121PP2018	379/1722	127/573	Missense variant C379T Arg127Cys	*tar*S	Poly(ribitol-phosphate) β-N-acetylglucosaminyltransferase TarS
	-	-	Intergenic region C1550T	5SrRNA	5S ribosomal RNA (partial)

**Table 9 pathogens-13-00839-t009:** SNP detection in mutants derived from *E. coli* 095PP2016. NT_POS—position in the mutated gene. AA_POS—position of changed amino acids. del—deletion, *—stop codon.

Bacterial Mutants	NT_POS	AA_POS	Mutation Effect	Gene	Product
235PP2017	603/1104	201/367	Frameshift variant 603–606 del CACT Thr202fs	*omp*C	outer membrane porin C
1304PP2022	447/1104	149/367	Frameshift variant 447–454 del CGCGACCT Phe149fs	*omp*C	outer membrane porin C
	527/978	176/325	Frameshift variant 527–537 del AAAACTTGCAG Lys176f	*wzz*B	regulator of length of O-antigen component of lipopolysaccharide chains
1306PP2022	511/1104	171/367	Stop gained C511T Gln171 *	*omp*C	outer membrane porin C
	453/978	151/325	Stop gained T453A Tyr151 *	*wzz*B	regulator of length of O-antigen component of lipopolysaccharide chains
265PP2018	97/1041	66/346	Frameshift variant 193–196 dup CAGG Val66fs	*omp*A	outer membrane protein 3a (II *;G;d)
	303/789	101/262	Frameshift variant 303 del A Lys101fs	*wbb*D	UDP-Gal:alpha-D-GlcNAc-diphosphoundecaprenol beta-1,3-galactosyltransferase

**Table 10 pathogens-13-00839-t010:** Storage stability of anti-mastitis bacteriophage cocktail at 2–10 °C.

Time [Month]	Titer for Each Component of the Cocktail [PFU/mL]		Final Titer [PFU/mL]	Stability [% Log]
0	*E. coli*	2.16 × 10^8^	3.44 × 10^8^	100
	*S. aureus*	1.28 × 10^8^		
3	*E. coli*	1.22 × 10^8^	1.90 × 10^8^	97
	*S. aureus*	6.82 × 10^7^		
6	*E. coli*	1.25 × 10^8^	1.98 × 10^8^	97
	*S. aureus*	7.31 × 10^7^		
9	*E. coli*	1.47 × 10^8^	2.30 × 10^8^	98
	*S. aureus*	8.30 × 10^7^		
12	*E. coli*	9.16 × 10^7^	1.36 × 10^8^	95
	*S. aureus*	4.42 × 10^7^		
15	*E. coli*	7.26 × 10^7^	1.02 × 10^8^	94
	*S. aureus*	2.97 × 10^7^		
18	*E. coli*	1.12 × 10^8^	1.46 × 10^8^	96
	*S. aureus*	3.43 × 10^7^		
21	*E. coli*	1.04 × 10^8^	1.35 × 10^8^	95
	*S. aureus*	3.09 × 10^7^		
24	*E. coli*	1.04 × 10^8^	1.38 × 10^8^	95
	*S. aureus*	3.42 × 10^7^		

**Table 11 pathogens-13-00839-t011:** The summary of phage receptor prediction.

Bacteriophage	Phage Genus	Receptor Predicted from Phage Genome	Receptor Prediction Confirmation by Phage-Resistant Mutants (Mutated Genes)
303Ecol101PP	*Tequatrovirus*	LPS core and OmpC	**235PP2017, 1304PP2022, 1306PP2022** (*omp*C, *wzz*B),
310Ecol104PP		LPS core and OmpA and PhoE	**265PP2018** (*omp*A, *wbb*D)
308Ecol101PP	*Mosigvirus*	LPS core and OmpC	**235PP2017** (*omp*C, *wzz*B)
348Ecol098PP		LPS core and OmpC or OmpF or maltoporin	**265PP2018** (*omp*A, *wbb*D)
241Ecol014PP	*Vequintavirus*	Lipopolysaccharide (sugar) and unknown membrane protein	No mutant sequences for analyses
351Saur083PP	*Rosenblumvirus*	β-O-N-acetylglucosamine of wall teichoic acid and supporting other polysaccharides	**119PP2018, 120PP2018,** **121PP2018** (*tar*S)
355Saur083PP	*Kayvirus*	β-O-N-acetylglucosamine of wall teichoic acid and supporting other polysaccharides	**119PP2018, 120PP2018,** **121PP2018** (*tar*S)
357Saur119PP		Putative α-O-N-acetylglucosamine of wall teichoic acid and supporting other polysaccharides	No obtained mutant

## Data Availability

The original contributions presented in the study are included in the article/Appendix A; further inquiries can be directed to the corresponding author/s. The complete genomes of bacteriophages included in the article are available in GenBank at the following accession numbers: OR062943 (241Ecol014PP), OR062944 (303Ecol101PP), OR062945 (308Ecol101PP), OR062946 (310Ecol104PP), OR062947 (348Ecol098PP), OR062948 (351Saur083PP), OR062949 (355Saur083PP), and OR062950 (357Saur119PP).

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
