# Peer review of "A New Approach for Phage Cocktail Design in the Example of Anti-Mastitis Solution"

_pathogens, 2024, doi:10.3390/pathogens13100839_

Round 1

Reviewer 1 Report

Comments and Suggestions for Authors

The article is devoted to the important topic of creation of effective phage cocktail against pathogens causing mastitis in animals. In general, the work is performed at a high level, but there are deficiencies in it, which should be supplemented with information to obtain a complete picture of the results obtained.

Minor revisions:

Line 100: Please specify time of incubation

Lines 124-127: Please describe methods of phage lysates preparation and purification (if applicable). Please add information about phage cocktail properties: solution base (medium, saline etc.), if known its pH, purity etc.

Lines 132 and 187: Please describe used NGS method (in particularly method of library preparation)

Lines 134-135: Please specify versions of used software and environment.

Lines 139-141: Self developed pipelines should be somehow described or if applicable referred to used known parts of them in order to present used methodology. Otherwise results of detected structural proteins and receptor binding proteins have no justification and should not be taken into this manuscript.

Line 158: Please specify diluent.

Line 161: Is it phosphate buffered saline? Please specify pH of the buffer.

Line 162: Please specify used medium

Line 165: Please describe at first abbreviation OD

Line 166: What kind of results were used for the inhibition degree assumption? Were there preliminary tests? If yes, please describe them in Supplementary Materials.

Lines 198-199: What means abbreviation “+G”? Please specify agar percentage in used soft agar and solid agar.

Lines 201, 492: Please specify used stabilizing solution

Line 224: What glycine buffer? pH? DMSO producer?

Line 247: Please specify fat percentage of the milk after centrifugation

Line 265: What kind of double agar layer method was used? Whole plate for a single dilution or a spot test? Please specify used bacterial strains, media and conditions.

Major revisions:

Section 2.2. Please describe method used for assessment of phage concentration.

Section 2.7. Please add citation for the used method. If the method for biofilm eradication assessment with MTT is original, please reflect its features and limitations in the Discussion.

Section 3.7. It is not clear what an input make E. coli and S. aureus phages on the titer. Please provide detailed information regarding each cocktail component (an if possible about each phage).

Discussion section: Please describe more clearly necessity of phage 355Saur083PP for the cocktail. It seems to be not capable to overcome devepoled resistance. Are there any known issues for the phage 357Saur119PP (amplification, stability etc)?

Figures and Tables from the Discussion section should be moved to the Results or Supplementary materials. Figure 7 could be used as a graphic abstract if applicable.

Please add graphical information for detected mutations in bacterial strains/resistant variants or add raw NGS data.

Supplementary Materials have no open access and thus not available for the peer review process.

Reviewer 2 Report

Comments and Suggestions for Authors

The manuscript presents a novel approach to the design of phage cocktails aimed at treating mastitis, a common and economically significant condition in dairy animals. The study focuses on the development and evaluation of a phage cocktail targeting Escherichia coli and Staphylococcus aureus, two prevalent pathogens in mastitis. The authors employ a detailed bioinformatic characterization of the phages, followed by in vitro testing to assess the efficacy of the cocktail in reducing bacterial load and preventing biofilm formation. The manuscript introduces a rational, bioinformatic-based method for designing phage cocktails, which could enhance the effectiveness of phage therapy for mastitis.

There are few important points that authors should add to the manuscript

Introduction:

Please add more information related to the alterantives to antimicrobials using referencis such as following:

    Tomanić, D.; Kladar, N.; Radinović, M.; Stančić, I.; Erdeljan, M.; Stanojević, J.; Galić, I.; Bijelić, K.; Kovačević, Z. Intramammary Ethno-Veterinary Formulation in Bovine Mastitis Treatment for Optimization of Antibiotic Use. Pathogens 2023, 12, 259. https://doi.org/10.3390/pathogens12020259

Antimicrobial use represent important driver for antimicrobial resistance development indicating the importance of development of antimicrobial alternatives. Hence, plaese references that include that relatinoship such as following:

    Kovačević, Z.; Samardžija, M.; Horvat, O.; Tomanić, D.; Radinović, M.; Bijelić, K.; Vukomanović, A.G.; Kladar, N. Is There a Relationship between Antimicrobial Use and Antibiotic Resistance of the Most Common Mastitis Pathogens in Dairy Cows? Antibiotics 2023, 12, 3. https://doi.org/10.3390/antibiotics12010003).

Conclusion:

Please incorporate if you have a plan to conduct in future in vivo studies to validate the effectiveness and safety of the phage cocktail in a practical setting. This would provide a more comprehensive assessment of its potential as a therapeutic option.

Overall, the manuscript presents a valuable contribution to the field of phage therapy, particularly in the context of mastitis treatment. The innovative approach to cocktail design and the promising in vitro results are commendable. However, to fully establish the practical utility of the phage cocktail, further studies, including in vivo evaluations and mechanistic investigations, are necessary. I recommend the manuscript for publication, provided that the authors address the aforementioned concerns in a revised version.

Reviewer 3 Report

Comments and Suggestions for Authors

The authors aimed to describe the use of bacetriophages in the treatment of mastitis.

In general, I liked the idea and the manuscript and I support publication. Nevertheless,, herebelow, some points that require improvement to bring the manuscript to publication standards.

1.      Please express the objectives of the study clearly and concisely.

2.      The omission of including streptococcal isolates among the test strains is a weak point of the work. This must be justified carefully, because one may think that the approach is ineffective against streptococci, which would reduce the value of the work.

3.      Please add a table with all the bacterial isolates used, in order to present their clinical details (year of isolation, source, type of infection etc.).

4.      Please add a separate section where to present all the controls used in the experimental work, e.g., consumables.

5.      Tables in results are very helpful for readers.

6.      In contrast, figures are of very quality and must be improved. Really, these figures are not acceptable.

7.      Discussion. 1) will you please calculate the expected annual reduction in the amount of antibiotics used against mastitis? 2) in clinical settings, how do you expect to administer the product to cattle? You need to provide some information or at least some ideas.

8.      References are OK and compatible with the text.

9.      Conclusion. Please dot add new ideas in this section, but rather do mention these in the Discussion. The section must be separated in two paragraphs.

Overall. A nice manuscript that can be accepted after making the above changes.

Round 2

Reviewer 1 Report

Comments and Suggestions for Authors

The authors of the manuscript "A new approach for phage cocktail design on the example of anti-mastitis solution" have significantly improved it. All minor and major revisions have been answered correctly. 

In my opinion, the manuscript is acceptable for publication after the following minor revisions:

Line 137: Specify the methods of biomass filtration (and other purification steps, if applicable) and microbial burden/sterility tests performed.

Line 166: Add citation for the SPAdes assembler.

Line 259: Specify correct citation number.

Line 292: Specify origin of the milk sample (included animal species and if known strain) and if applicable method used for the fat control.

Line 293: Remove decimal point from the rpms. Add rotor type (swinging bucket or fixed angle). Add RCF (or g) or rotor radius (or diameter).

Reviewer 3 Report

Comments and Suggestions for Authors

All issues were addressed. No further comments.

Author Response

Dear Reviewer, 

Once again, we sincerely appreciate your valuable comments and insights.

Yours faithfully, 

Daria Królikowska 

Agnieszka Kajdanek